# Replicating community dynamics reveals how initial composition shapes the functional outcomes of bacterial communities

A. Pascual-García [1,2,6], D. W. Rivett [3,6], Matt Lloyd Jones [4] & T. Bell [5] ✉

Bacterial communities play key roles in global biogeochemical cycles, industry, agriculture, human health, and animal husbandry. There is therefore great interest in understanding bacterial community dynamics so that they can be controlled and engineered to optimise ecosystem services. We assess the reproducibility and predictability of bacterial community dynamics by creating a frozen archive of hundreds of naturally-occurring bacterial communities that we repeatedly revive and track in a standardised, complex resource environment. Replicate communities follow reproducible trajectories and the community dynamics closely map to ecosystem functioning. However, even under standardised conditions, the communities exhibit tipping-points, where small differences in initial community composition create divergent compositional and functional outcomes. The predictability of community trajectories therefore requires detailed knowledge of rugged compositional landscapes where ecosystem properties are not the inevitable result of prevailing environmental conditions but can be tilted toward different outcomes depending on the initial community composition. Our results shed light on the relationship between composition and function, opening new avenues to understand the feasibility and limitations of function prediction in complex microbial communities.

Bacterial communities are complex systems whose dynamics are difficult to predict. Even if individual populations are well-studied, predicting the trajectories of diverse, natural bacterial communities is challenging because of in-built contingencies; a slight, stochastic increase in the abundance of one population could have cascading impacts that steer the community along a divergent trajectory. Bacterial community dynamics may be broadly predictable in simple environments[1–3] but usually only at a coarse level of taxonomic resolution[4,5], likely because taxa that are closely related also occupy similar ecological niches[6]. Synthetic consortia comprising well-characterised or genetically modified strains exhibit predictable dynamics[7] that in some cases can be used to control proscribed ecosystem properties[8] or confer phenotypes to a host[9,10]. However, natural communities are much more diverse, typically containing thousands of interdependent taxa that compete for resources, exchange metabolites, and exhibit sophisticated coordinated behaviours like quorum

[1]Centro Nacional de Biotecnología, CSIC, Madrid, Spain. [2]Institute of Integrative Biology, ETH, Zürich, Switzerland. [3]Department of Natural Sciences, Faculty of Science and Engineering, Manchester Metropolitan University, Manchester, UK. [4]European Centre for Environment and Human Health, University of Exeter, Penryn, UK. [5]Imperial College London, Silwood Park Campus, Ascot, UK. [6]These authors contributed equally: A. Pascual-García, D. W. Rivett. ✉e-mail: thomas.bell@imperial.ac.uk

sensing. Furthermore, high levels of niche overlap among taxa (functional redundancy) can lead to a lottery for community membership and therefore to community dynamics that are governed by chance colonisation order[11].

These observations and ideas make two, apparently contradictory, predictions. The first idea predicts *convergent* community trajectories under standardised environmental conditions. If bacterial communities contain high levels of diversity, natural selection would be expected to rapidly and reproducibly sort the best-adapted taxa, resulting in a single taxonomic and functional outcome. This idea is supported by studies showing a predictable simplification and convergence of communities in standardised conditions[4]. Under this scenario, community composition seems to go hand-in-hand with community-level functioning, and therefore it may be possible to accurately predict functioning without a detailed knowledge of the mechanistic details underlying microbial dynamics, as illustrated by the functional landscape approach[12]. The second idea predicts *divergent* community trajectories, generating a rugged compositional landscape that may or may not reproducibly achieve specified functions. This idea is further supported by observations and theory showing that alternative community states arise across many study systems, perhaps facilitated by flexible decision-making in bacterial resource acquisition[13]. Alternative compositional states could have a range of consequences for ecosystem functioning if the taxa making up the alternative compositional states have different metabolic repertoires; a result that may have been overlooked due to the species-poor communities explored in previous studies (typically lower than 20, see e.g.[14,15]). However, alternative compositional states could have the same functionality if many bacteria have overlapping functional capacities (redundancy), resulting in the same ecological niches being filled. This has been implied by studies that have shown taxonomic

divergence but functional convergence[16]. Finally, more complex relationships are also plausible; for example, high-order interactions may dominate the relationship between composition and functioning[17].

To address these conflicting ideas, we considered a large set of complex communities to re-frame a famous question from evolutionary biology[18]: Does replaying the tape of ecology produce the same compositional and functional outcome?[19].

Here, we run the tape of ecology 4 times for 275 complex communities containing hundreds of different taxa to identify principles of bacterial community dynamics. We consider the fate of communities with different initial taxonomic compositions inoculated into a standardised, complex resource, sterile environment. Each starting community is a naturally occurring community of heterotrophic bacteria involved in the degradation of beech (*Fagus sylvatica*) leaf litter in miniature ponds. Leaf litter degradation is an important ecosystem process because increased degradation results in more rapid biogeochemical cycling thereby increasing the productivity of the ecosystem, while lower degradation results in greater carbon storage. Previous experiments have shown that these communities exhibit a strong relationship between degradation rates and the diversity[20] and taxonomic composition[21] of the communities. This system therefore offers a tractable avenue for studying the ecology of natural bacterial communities associated with the provision of a particular ecosystem service. Source microbiomes are taken from 275 rainwater pools from the buttressing of beech trees, the bacterial community is separated from co-occurring biota and from the surrounding environmental matrix, and the whole bacterial communities are cryopreserved. The frozen communities are revived independently four times, re-grown repeatedly in a standardised microcosm containing a sterile beech leaf-based growth medium (see Fig. 1a). We quantify the taxonomic composition of the communities before they were revived from cryopreservation

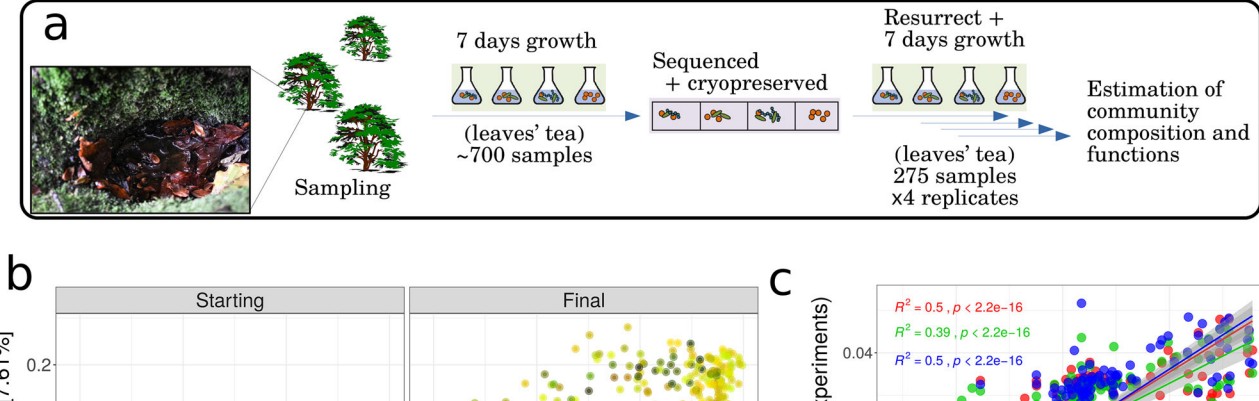

**Fig. 1 | Initial states predict final states. a** 275 samples obtained from rain pools were cryopreserved, revived, and inoculated into a standardised growth medium. Community composition was assessed using amplicon sequencing at the start (before cryopreservation) and end of the experiment (revived communities). **b** Principal coordinates analysis (PCoA) of all communities with colours representing their position in ordination space at the start of the experiment. The left panel shows the starting communities and the right panel shows the final communities. The third PCoA component is shown in Supplementary Fig. 1. **c** Composition of final communities in the high-dimensional space was predicted as a rigid-

body transformation of starting communities. The transformation was obtained by applying a singular value decomposition of the cross-covariance matrix of the starting communities and one replicate of the final communities (see "Methods"). The figure shows a linear relationship between the first SVD component of transformed starting communities (prediction) and the final communities not used to find the transformation (experiments) (second SVD component in Supplementary Fig. 2). The Pearson $R^2$ and two-sided *t*-statistic *p* values are independently indicated for each final replicate, with shadowed regions indicating 95% CI. Source data are provided as a Source Data file.

(starting communities) and their composition and functioning at the end of the experiment (final communities), allowing us to understand their community trajectory, the reproducibility of those trajectories, and how composition and functioning are related.

## Results and discussion

### Replicate communities have predictable trajectories

We first identified whether the 4 revived replicates of each of the 275 cryopreserved communities were clustered at the end of the experiment. We performed a signal-to-noise analysis by quantifying the ANOSIM R statistic[22], which ranges between zero (random groupings) and one (distinct groups). The four replicates produced remarkably non-random groupings ($R = 0.716$, 275 groups, $p$ value $< 10^{-3}$) and there was no significant difference among the replicates ($R = 0.004$, 4 groups, see Supplementary Table 1), hence ruling out a stochastic assembly. The reproducibility of the community trajectories was visualised by tracking changes in the relative abundance of the hundreds of taxa underpinning taxonomic changes using ordination. The ordination revealed that the direction of the shift from starting- to final communities was remarkably consistent among communities and replicates (Fig. 1b and Supplementary Fig. 1). To test this idea by formally considering the whole multidimensional space, we asked whether the final composition of one of the replicates could be obtained using a rigid-body transformation (i.e. translation and rotation) of the starting communities. We first showed that there was a small and significant Root Mean Square Deviation between the transformed starting communities and one replicate of the final community (0.48, randomised 95% C.I. [0.54, 0.56], see "Methods"). We then asked whether the transformed starting communities predicted the composition of the three remaining replicates by comparing the main components of a singular value decomposition (SVD) of the transformed communities against the SVD of each of the three remaining replicates. We found a strong and significant linear relation (Fig. 1c and Supplementary Fig. 2), confirming that a linear transformation leads to an accurate prediction of the composition of resurrected communities and therefore that groups of taxa collectively move in similar directions in the compositional space. The finding appears to reject the 'divergence' hypothesis since even communities that were independently revived from cryopreservation had predictable trajectories- once the trajectory of one replicate was known, the community could be repeatedly revived to produce the same outcome. However, as we will show immediately, compositionally similar communities do not always show similar trajectories between them.

### Predictability of community trajectories depends on initial composition

At the start of the experiment, the communities were not uniformly distributed across compositional space[23]. To refine our analyses, we, therefore, performed unsupervised clustering of the communities to identify compositionally similar sets of communities termed classes, similar to the concept of gut microbiome enterotypes[24]. Clustering was performed considering an all-against-all Jensen–Shannon distance matrix[25] of the communities' relative abundances in the multidimensional space (see "Methods" and Supplementary Note 1). Intuitively, community classes can be regarded as attractors in the compositional landscape towards which the composition tends to gravitate. Community attractors under a specific set of environmental conditions can best be identified by tracking the trajectory of communities originating from multiple compositional starting points, which is what we have done here. We used the ANOSIM R statistic to measure how sharply classes are delimited, allowing us to identify an absolute maximum of 17 starting community classes (ANOSIM $R = 0.68$, $p$ value $< 10^{-3}$) and a second maximum with 5 local classes ($R = 0.64$, $p$ value $< 10^{-3}$). For simplicity and consistency with previous work[23], we analysed the five classes (Fig. 2; the PCoA was used

for visualisation only whereas clusters were determined as indicated above).

In previous work, the classification identified six classes (with representatives of one of them missing in the selection of communities made in this work). Approximately 50% of the pairs of communities found in the same cluster in the previous work also clustered together here. Discrepancies were because the previous work identified Operational Taxonomic Units whereas the current analysis used Amplicon Sequence Variants.

Importantly, analysis of classes in previous work[23] showed that they had well-differentiated metagenomic and functional signatures, providing strong evidence of selection. We speculated that the starting bacterial communities would have experienced a range of environmental conditions due to site-specific differences in leaf inputs, precipitation, oxygen availability, and many other factors, resulting in a relatively diverse array of community classes[23], later confirmed by a relationship between composition and the size of the tree-holes, which may have, in turn, an influence on the variables listed[26]. In this work, the communities retained signatures of their provenance despite being re-grown twice on the beech leaf media (they were first grown to stationary phase before cryopreservation and then second revived and grown across four replicates), with communities clustering in compositional space according to their collection location and date (Supplementary Fig. 3). As we will show below, they also showed well-differentiated metagenomic and functional signatures, suggesting that, by working with this growth medium, we captured the signal both of history and selection (from their native environment and from the medium itself). These observations suggest that these classes represent robust outcomes.

We detected two community classes at the conclusion of the experiment (ANOSIM $R = 0.78$, 2 groups, $p$ value $< 10^{-3}$) (Fig. 2b, vertical bars, and Supplementary Fig. 4), consistent with the idea that the standardised environmental conditions across the microcosms selected a more limited set of communities that had specific adaptations to the microcosm environment, and demonstrating that separating communities into classes was an economical description of the outcome of the experiment. This conclusion is further illustrated by the observation that 80% of communities had all four replicates ending in the same final class, and only 2.5% of the communities had replicates evenly split into the two final community classes. The outcome of each community was therefore highly dependent on the starting community: once the initial composition was known, its fate was ordained with only minor deviations. However, the specific class to which revived communities converged depended on the starting class: Starting Classes 1, 2 and 4 tended to end up in the same final class, while Starting Classes 3 and 5 were more unpredictable.

Therefore, although individual communities had remarkably reproducible trajectories, the trajectory of each community was contingent on the initial community class. We illustrate this contingency by showing the fate of each of the five starting community classes. Starting Class 1 and Class 4 communities consistently converged toward Final Class 1 communities, with ~90% of the starting communities originating from those classes (Fig. 2). A hypothetical study that used one or a few communities that were within either of those classes would have observed a single outcome, consistent with the idea that communities converge and simplify under standard conditions[4]. Other communities were less predictable in their trajectories at the class level, with outcomes for communities in Starting Classes 2, 3, and 5 more evenly split between Final Class 1 (~20–50%) and Final Class 2 (~50–80%) final communities (Fig. 2). A study that used one or a few of the communities from starting community Classes 2, 3, or 5 would therefore likely have observed the formation of alternative compositionally and functionally divergent outcomes, thus diminishing the predictability of trajectories.

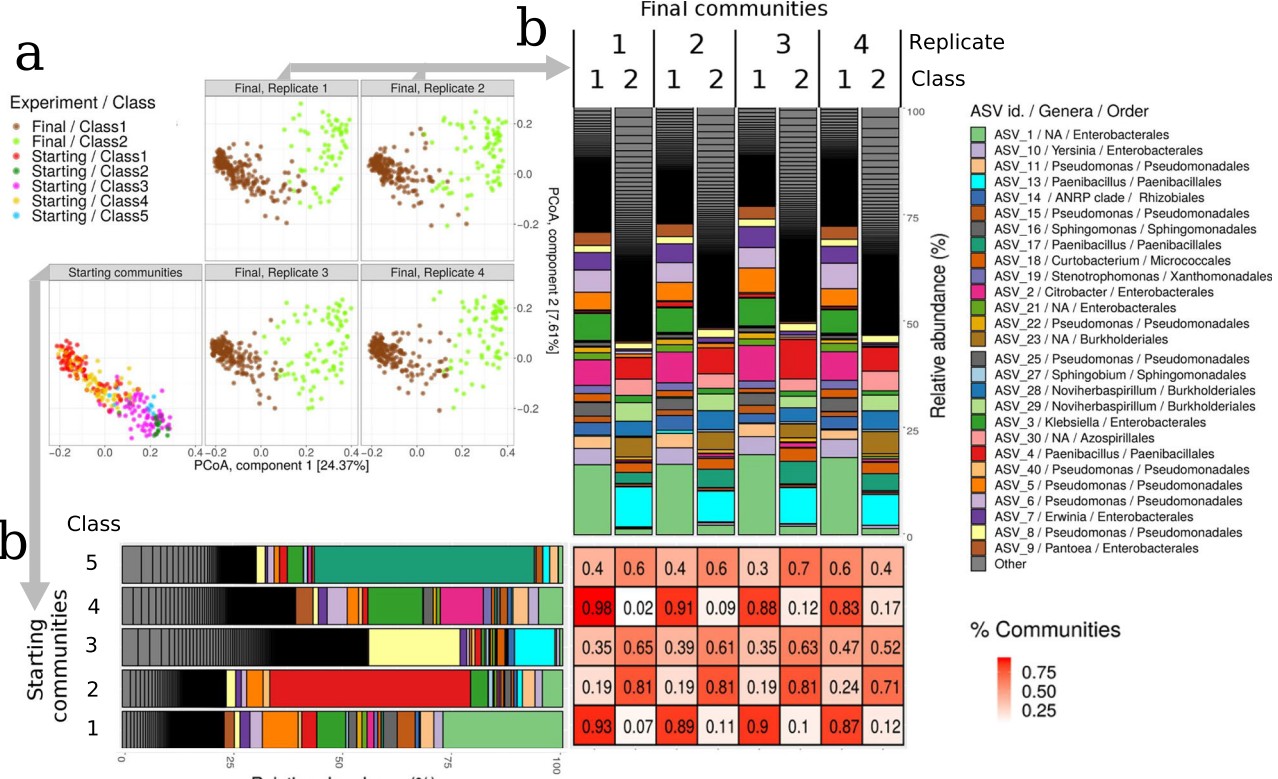

**Fig. 2 | Compositional convergence. a** Principal coordinates analysis (PCoA) of all communities (as in Fig. 1b) with colours representing the starting and final community classes (compositionally similar clusters of communities determined by unsupervised clustering). The replicates of the final communities were separated into different panels for clarity to show the reproducible outcome. **b** Bar plots showing the relative abundance of Amplicon Sequence Variants (ASVs) for starting communities (horizontal bars) and final communities (vertical bars). The communities were divided into the five starting community classes and the two final community classes. The final communities were further divided into the four replicates. The 20 most abundant ASVs were identified into genus and the remainder were combined into 'Other'. The taxa in the bars follow the order indicated in the legend. Genera that were not resolved are indicated with NA. The matrix indicates the proportion of each starting community class (rows) that resulted in each final community class (columns). Source data are provided as a Source Data file.

## From tape to landscape

To understand previous results we propose a conceptual picture in which the experimental communities can be visualised as traversing a compositional and functional landscape (Fig. 3a), with communities starting from compositions set by their native environmental conditions, and where community stability across the landscape is determined by the new environmental conditions. Stable communities would be those that are resistant to change and would return to their original composition if they are perturbed, while unstable communities would rapidly traverse compositional space until they reach neighbourhoods where the communities are more stable. If there was a single stable composition, any starting community composition would generally converge on this single attractor[23]. If there are several attractors, communities located near saddle points or on ridges of the landscape would have a more unpredictable outcome since initial small changes to the composition would tilt the communities toward alternative attractors. The topography of the landscape determines the predictability and repeatability of the community dynamics and therefore provides information about the reproducibility of community trajectories (see Supplementary Note 2 and Supplementary Fig. 5).

The experimental results are consistent with a rugged landscape containing at least two attractors toward which the composition tended to gravitate (i.e. Fig. 3b). Starting community Classes 1 and 4 lie along the flank of the attractor close to Final Class 1, resulting in convergence to a single outcome (Fig. 3b and Supplementary Fig. 6). By contrast, starting Classes 2, 3 and 5 straddle a ridge that allows them to diverge to Final Class 1 or 2 (Fig. 3b and Supplementary Fig. 6).

To quantify this idea we computed the distance of each starting community to the centroids and to the borders of the final attractors (Fig. 3c). For a given starting community, the border was considered to be the closest community associated with each final attractor. Communities that were further away from both centroids tended to converge to Final Class 2 particularly those starting communities that were closer to both its centroid and border. The Final Class 2 attractor is therefore less steep or less accessible from the starting community compositions. Starting communities needed to be far from the final community Class 1 attractor to avoid its pull, and would only converge to Final Class 2 if they were sufficiently distant.

This was further confirmed when we computed the mean distance between starting and final classes, with Starting Classes 1 and 4 (the largest set of communities) having a high similarity with Final Class 1, suggesting that these two starting classes were already orbiting an attractor that, after the second round of growth, consolidated into a single large and robust attractor (Supplementary Fig. 7). By contrast, Starting Classes 2, 3 and 5 were more similar to Final Class 2, but their mean similarity was much lower, suggesting a substantial compositional transformation of the communities falling into this attractor (Supplementary Fig. 7).

As illustrated here, rugged landscapes may exhibit convergent or divergent outcomes depending on the starting location in compositional space. A detailed knowledge of these rugged compositional landscapes would be needed to avoid the risk of diverging to potentially undesirable compositional and functional outcomes.

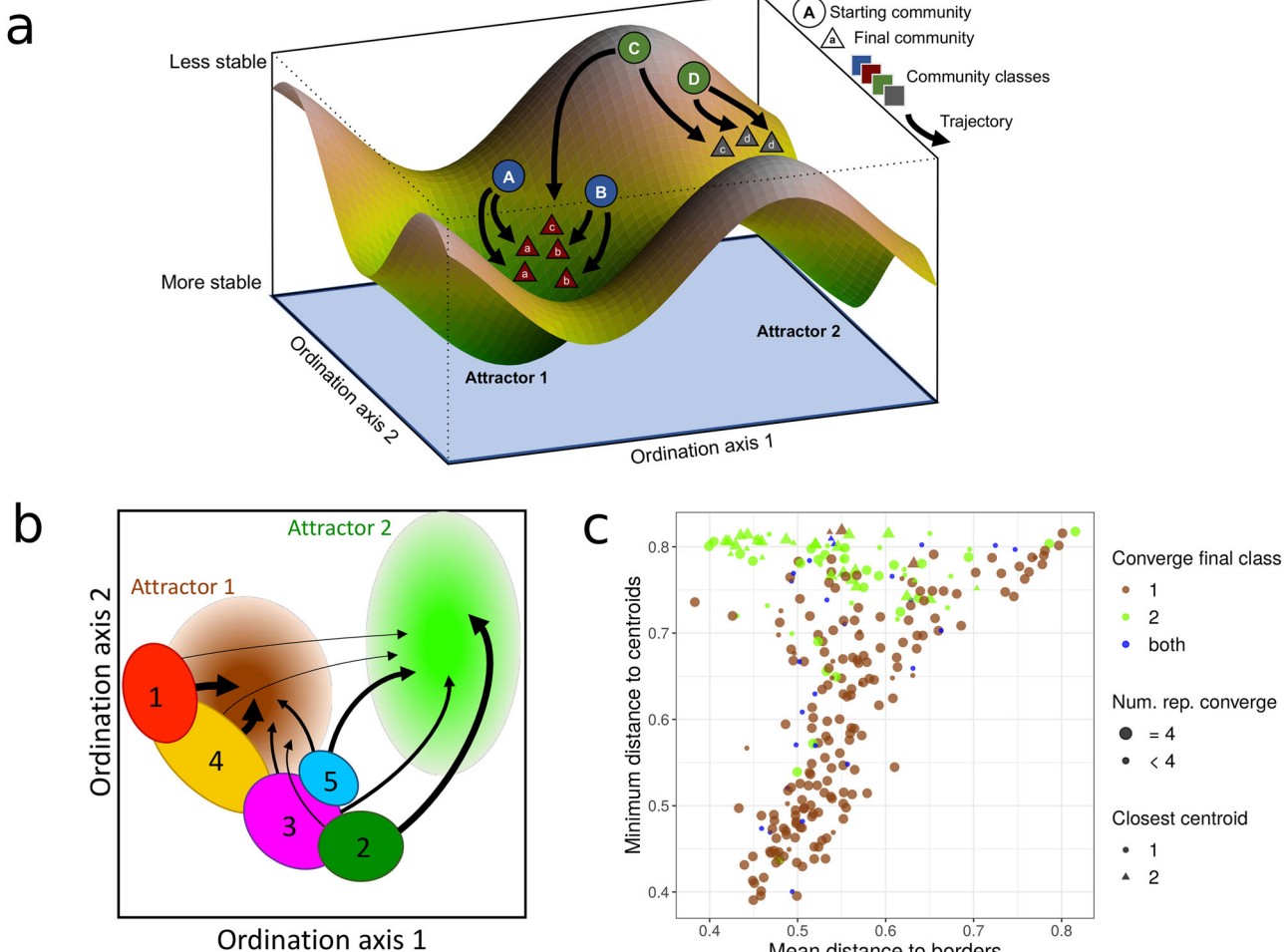

**Fig. 3 | Community trajectories are influenced by the topography of the compositional landscape. a** Illustration of the compositional landscape from the perspective of a set of starting communities entering a new environment. Two attractors are shown, with circles representing the starting communities, triangles representing the final communities, shape colours indicating community classes, and arrows indicating community trajectories. **b** Illustration of the location (in ordination space) and trajectory of each starting class used in the experiment. Ovals indicate approximate locations of the numbered starting classes, arrows indicate trajectories with thicker arrows indicating more consistent outcomes, and pale circles indicate putative attractors. An equivalent representation with real data is provided in Supplementary Fig. 6. **c** For each starting community, we computed the minimum distance to the centroid of the final community classes and the mean distance to the borders. The border was defined as the closest community in the final class. We observed that starting communities that were distant from both centroids converged to Final Class 2 particularly when they were closer to the borders of both attractors. Source data are provided as a Source Data file.

## Core sets of ASVs determined convergence towards the attractors

In previous sections, we observed that the positions of the starting communities in the compositional landscape determined the trajectory of the communities. Since positions are given by the presence-absence of taxa and their relative abundances, we asked if it is possible to identify ASVs, or groups of ASVs, that characterise the different trajectories (see "Methods"). To this end, we classified the community trajectories as 'convergent' if all four revived replicates from the same starting community converged to the same final class, or as 'divergent' otherwise. Therefore, for each trajectory, one starting community and four final communities were labelled as convergent (to Class 1 or 2) or divergent. Next, we studied whether each ASV tended to have a propensity to be observed more often in a particular type of trajectory. To assess this question we estimated the statistical propensity of each ASV to be observed in communities belonging to convergent or divergent trajectories (see Fig. 4a and "Methods"). The statistical propensity is the log-fold change in the probability of observing an ASV in a given trajectory compared to the probability of observing that ASV in any trajectory. A positive association between an ASV and a trajectory was found when the ASV propensity for that trajectory was significantly

positive (i.e. 95% bootstrapped propensities were positive, see "Methods"). Propensities were independently computed for starting and final communities, which allowed us to identify whether ASVs maintained (or not) their affinity to specific trajectories. An overview of the association between ASVs and trajectories for the most represented families is shown in Fig. 4b.

ASVs that had a positive and significant propensity to be in communities that followed convergent trajectories mostly ended in Class 1 (45% of those with propensities for convergent trajectories), while only 1.4% ended in Class 2. Notably, we identified a group of ASVs with a propensity towards communities in trajectories converging to Class 1 (in both starting and final experiments, hereafter group 'S&F1'). This group represented a large fraction of the relative abundances in communities belonging to this type of trajectory (see Fig. 4c): median relative abundances of S&F1 members were above 0.4 for communities in trajectories converging to Class 1, while they were below 0.020 for those converging to Class 2. Conversely, we found another group of ASVs with a significant propensity for being observed in communities of the three types of trajectories in both starting and final experiments ('cosmopolitan' group) whose abundances, however, tended to be high only in Class 2 communities. Their median relative abundances were above 0.18 for

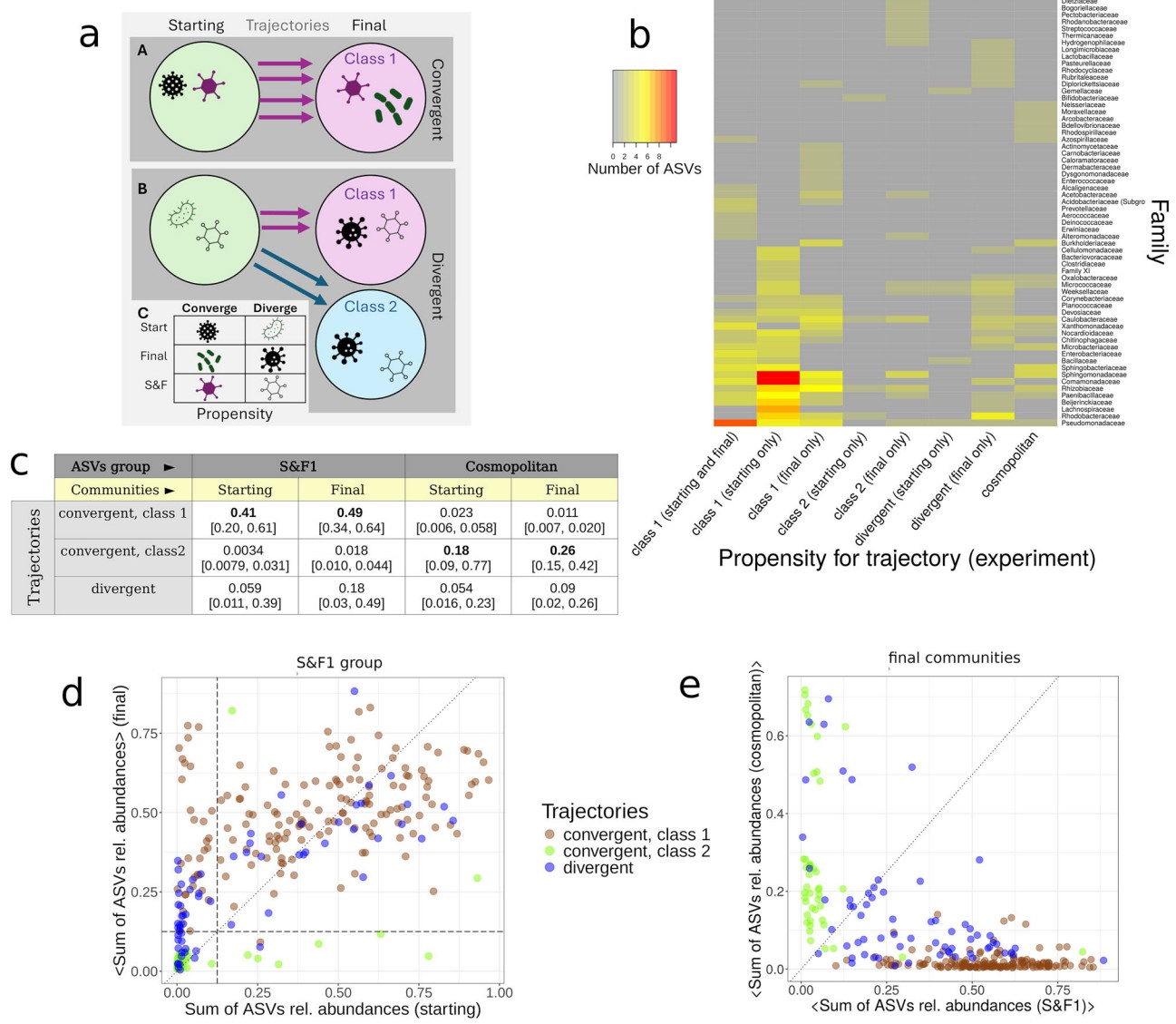

**Fig. 4 | Associations between taxa and convergence. a** Communities (circles) at the start and end are first categorised according to whether they are (A) convergent, where all four replicates (arrows) result in the same final class or (B) divergent, where at least one replicate ends in a different final class. The shapes within the communities represent different taxa. (C) We quantified the propensity for taxa to be associated with convergent or divergent trajectories. Taxa are listed according to whether they have a propensity to be associated with communities that converge to Class 1 (Converge) or diverge to Class 1 and 2 (Diverge). Taxa were further categorised according to whether these associations arose due to their presence in the Starting, Final, or both starting and final (S& F) communities. Taxa in the same categories belonged to the same propensity group. **b** Number of ASVs per family found in some relevant propensity groups. Only the top 25 families per group were shown. **c** Median [interquartile values] of the sum of the relative abundances of S& F1 and cosmopolitan groups in the starting and final ASV tables (columns) for communities classified in each type of trajectory (rows). **d** Sum of the relative abundances of ASVs belonging to the S& F1 propensity group for the starting vs. final experiments (averaged across replicates). **e** Sum of the relative abundances of ASVs belonging to the S& F1 (x-axis) and cosmopolitan (y-axis) groups in the final experiment (averaged across replicates). Source data are provided as a Source Data file.

communities in trajectories converging to Class 2, while median values were below 0.02 for those converging to Class 1. Therefore, high relative abundances of S&F1 and cosmopolitan groups were indicative of trajectories converging to Classes 1 and 2, respectively.

To explore these results further, we examined changes in the summed relative abundances of S&F1 members between starting and final experiments (the latter averaged across replicates) for all communities (Fig. 4d). Each community was coloured according to its trajectory type. We found that if the summed relative abundance exceeded ~0.125 the community almost always converged to Class 1 (Fig. 4d, vertical dotted line), and those converging to Class 2 had their final values fell below this threshold with just two exceptions (Fig. 4d, horizontal dotted line). Therefore, these observations, with rare exceptions, allowed us to identify a tipping point determined by a core

group of co-selected ASVs that led to convergence to the most populated attractor (Class 1).

The inverse result was observed for the cosmopolitan ASVs group, with communities in Class 1 trajectories falling below a threshold of -0.2 (see Supplementary Note 3 and Supplementary Fig. 8) and communities in Class 2 trajectories above it. Combining both results, we observed that by representing the sum of the relative abundances of S&F1 vs. cosmopolitan group members it was possible to clearly separate communities belonging to both types of convergent trajectories (Fig. 4e). Importantly, communities in divergent trajectories often had their relative abundances more evenly distributed among both groups, indicating that unbalanced starting conditions between these two groups are necessary to ensure predictability in the trajectories.

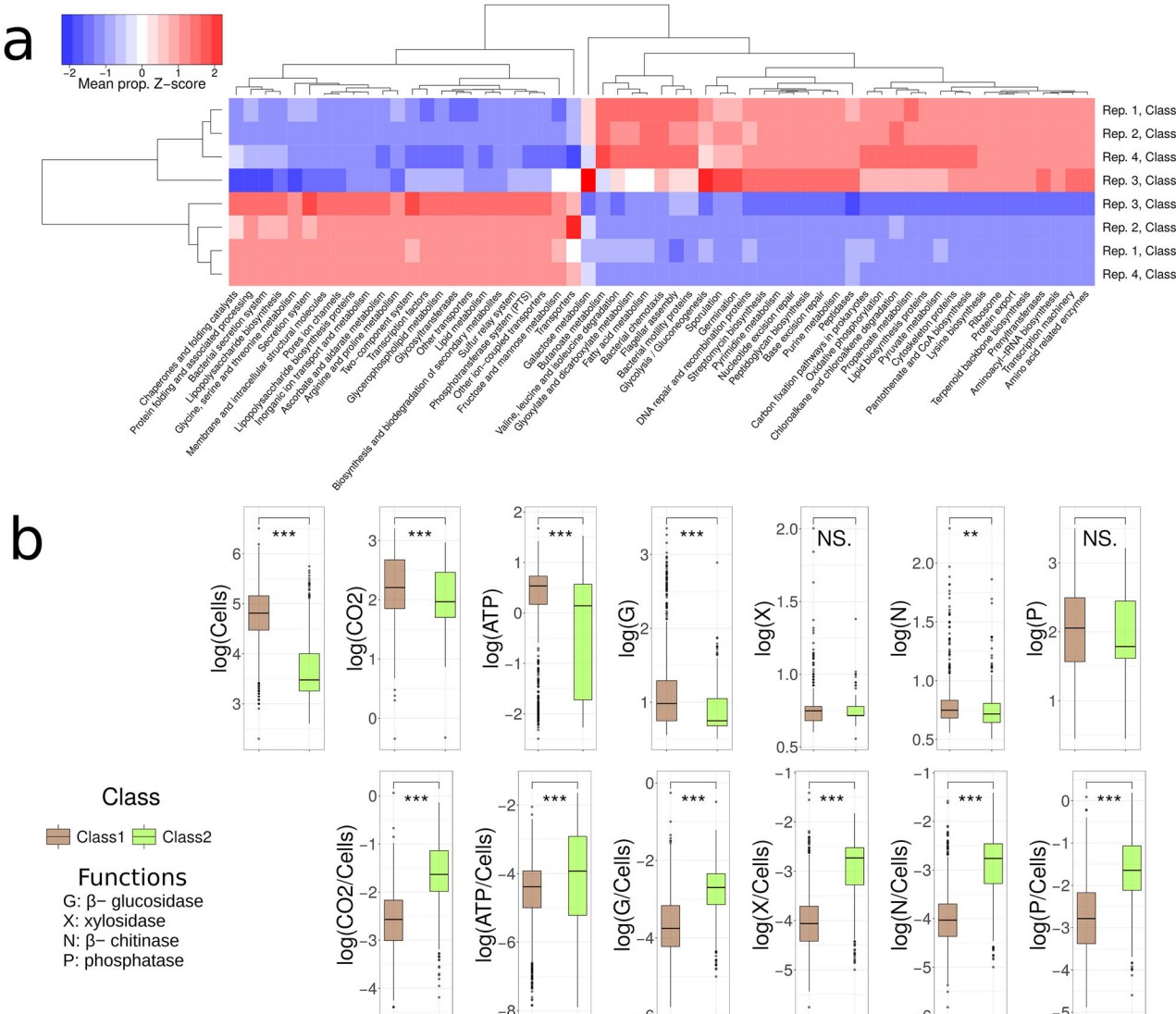

**Fig. 5 | Functional convergence. a** *Z*-score of the mean proportion of genes clustered in KEGG metabolic pathways found in final community classes for each replicate (Rep). The scaling of the *Z*-score was computed for each pathway (i.e. scaled by columns). Only pathways showing a significant difference between at least two classes are shown. **b** The functional measurements were significantly different between the two final community classes. Boxes and horizontal lines represent the quartile boundaries, and upper and lower whiskers extend $1.5 \times$ IQR from the correspondent hinge. Outliers are also shown. Biological replicates belonging to equivalent classes were merged, thus representing 794 communities in Class 1 and 306 in Class 2. Two-sided Wilcoxon test, significance *p* value: \*\*\*0.001, \*\*0.01, NS not significant. Source data are provided as a Source Data file.

## Community classes reflect functional differences

Our results suggest that an in-depth exploration of the compositional landscape may allow us to select communities that will follow specific trajectories and, more specifically, that may end up in specific attractors. We then asked whether these attractors were functionally equivalent or whether they showed distinctive functionality. We explored ways of mapping functional data onto the community trajectories by inferring the functional profiles of the communities from their composition and by direct functional measurements. We first predicted the metagenomes of the communities from the 16S rRNA sequencing data using PiCRUST to categorise genes by function using the KEGG database[27]. This analysis suggested that the 2 final community classes were associated with distinct modes of nutrient uptake. In Final Class 1, which contained more than two-thirds of final communities (Supplementary Table 1), there was a higher proportion of genes related to rapid nutrient uptake (e.g. transporters, phospho-transferase system), transcription factors, and fructose and mannose metabolism (see Fig. 5a, and Supplementary Figs. 9 and 10 in

Supplementary Note 4). There was also a higher proportion of genes associated with lipo-polysaccharide metabolism and membrane and intracellular structural proteins. By contrast, Final Class 2 communities had a higher proportion of genes related to oxidative phosphorylation and pathways related to the acquisition of acetyl-coA (e.g. degradation of amino acids, fatty acids metabolism, and propanoate metabolism), implying adaptions to resource-poor environments and a higher investment in resource acquisition and usage. Final Class 2 also retained pathways that were more abundant in starting Classes 2 and 5, including genes related to chemotaxis and motility and genes associated with hostile environments such as sporulation (Supplementary Fig. 11). Final Class 2 comprised less than one-third of all revived communities (Supplementary Table 1), implying that the majority of communities gravitated toward improved leaf medium uptake, while a minority may be relics of communities that were poorly adapted to the beech leaf culture media or became self-limiting (e.g. due to build-up of waste products)[23].

We validated the metagenomic predictions by measuring the degradation rate of four substrates along with the whole-community metabolic activity, respiration rate, and cell numbers in the final communities. Final Class 1 communities had higher degradation rates and enzymatic activity, while Final Class 2 communities had a higher per-cell investment but lower cell numbers, implying that they were growing inefficiently (Fig. 5b). We speculate that this was due to higher metabolic costs in this environment for tasks not associated with cell growth and division such as exo-enzymatic investment and that these communities may be adapted to environments with fewer or more recalcitrant resources. Both the metagenomic profiles and the functional experiments demonstrate that the the final community classes are not just alternative taxonomic variants providing the same functional outcome, but that the divergent community classes translated into significant functional differences. The two final classes are therefore unlikely to result from neutral community drift.

The experimental results demonstrate the reproducibility of community dynamics, a central requirement for understanding how microbial communities assemble and for establishing their utility as agents for altering ecosystems. Past efforts at engineering ecosystem services by manipulating or domesticating microbial communities have produced mixed results[28,29]. The experimental results provide a general explanation and route forward centred on characterising the topography of compositional and functional landscapes. By preserving and reviving hundreds of communities we were able to expand on this simple discovery. First, there was a strong historical contingency in the community trajectories, with naturally occurring compositional classes from the field predicting the compositional classes that communities gravitated towards in the laboratory. Second, these broad compositional classes also reflected the main axes of functional performance, both in terms of their functional capacity (metagenomes) and their delivery of ecosystem processes (degradation rates and activity). Statistical approaches aimed at estimating the functional landscape from composition[12] may fail for complex communities, such as those considered here. The reason is that the estimated functional landscape will exhibit less local ruggedness than the true landscape[30] unless alternative states in the compositional landscape are identified first. Third, while the community dynamics were highly predictable under standardised initial conditions, the trajectory depended on the initial community composition. Communities did not simplify to a single outcome, but individual communities may have different (predictable) trajectories. Together, the results show the feasibility of pushing ecosystems toward alternative compositional and functional outcomes by minor alterations to the community composition.

In this ecosystem, we identified two tipping points that determined the primary outcome (convergence to the attractors) using the summed relative abundances of two core sets of ASVs. These two simple thresholds predicted the outcome for many communities, although we note that the conditions imposed by these thresholds were not necessary (some communities converged that did not exceed the threshold) but they were almost sufficient (very few communities exceeding the threshold did not converge to the correspondent attractor). Moreover, having relative abundances above both thresholds seemed to be necessary to observe divergent trajectories. Therefore, we found necessary starting conditions affecting the predictability of the experiment. In addition, we suggest that some exceptions likely stem from ASVs beyond the core set. For example, some ASVs appeared to be repressed in communities associated with trajectories converging to Class 1 because their abundances were dramatically reduced by the end of the experiment, likely due to selection in the early stages of the experiment or in their native environment. This apparent repression was paralleled by a set of ASVs that increased from low relative abundances and that appeared to be co-selected with the Class 1 core set. These observations suggest a richer mechanistic process that appears to be orchestrated through the coordinated dynamics of sets of taxa,

rather than by individual keystone activities. Therefore, our results emphasise the importance of looking for levels of organisation between the community- and the 'species'-level to fully understand the dynamics of complex microbial communities[31].

## Methods

### Laboratory methods

We sampled 753 rainwater pools from the buttressing of beech trees (*Fagus sylvatica*) from August 2013 to April 2014[21]. The pools were stirred thoroughly to obtain an unbiased sample of the whole community and we collected 1 ml of water and sediment. The samples were diluted 1:4 in sterile phosphate-buffered saline (PBS, pH 7.0, Sigma-Aldrich) and filtered (pore size 20–22 μm, Whatman 4 filter paper) to remove debris. The filtrate containing the communities was inoculated into 5 ml sterile beech leaf medium (50 g dried beech leaves autoclaved in 500 ml PBS, filtered, diluted 32-fold in PBS, amended with 200 μg ml$^{-1}$ cyclohexamide (Sigma-Aldrich) to inhibit fungi). Each community was incubated at 22 °C under static conditions for 1 week to allow communities to reach the stationary phase. A sample was collected to characterise the (starting) community composition (16S rRNA sequencing, see "Sequencing methods") and communities were stored at −80 °C after the addition of glycerol, which acts as a cryoprotectant (final concentration 30% v/v glycerol, 0.85% w/v NaCl). Further details of the experimental methods on starting communities are provided elsewhere[21], which also includes documentation of how cryopreservation impacted the communities.

The experiment to revive and grow final communities was conducted in microcosms (1.2-ml-deep 96-well plates) containing 840 μl sterile beech leaf medium inoculated with 40 μl of each revived community (~20,000 cells) To ensure that there was no systematic bias in our experimental procedure, we thawed all of the communities at the same time for the experiments. The communities were thawed in this way on two separate occasions (each with two replicates) to conduct four independent trials of the experiment. Two hundred seventy-five communities were assayed, yielding a total of 1100 microcosms. The time between cryopreservation and resurrection was between 6 and 14 months depending on when the revived community was originally collected from the natural environment. The microcosms were incubated under static conditions at 22 °C for 7 days, after which time the communities were characterised for their taxonomic composition (see "Sequencing methods") and measured ecosystem functioning.

We measured bacterial activity, growth, and substrate degradation rates in microcosms. Measurements were generally taken at the end of the experiment except for respiration, which was measured cumulatively throughout the experiment. Final bacterial counts were obtained by staining the cells with thiazole orange (42 nM, Sigma-Aldrich) followed by obtaining absolute counts using a C6 Accuri flow cytometer (size threshold of 8000 forward scatter height (FSC-H)), with cells gated on the side scatter area (SSC-A) and fluorescence channel 1 (FL1-A) (533/30) channels. We used a threshold of 800 fluorescence units to distinguish cells from detritus. Bacterial respiration was measured using the MicroResp $CO_2$ detection system (www.microresp.com) according to the manufacturer's instructions, with absorbance readings converted to $CO_2$ using a standard curve[21]. Respiration measurements were taken as the cumulative respiration of the whole community over the 7-day incubation period. The potential for metabolic activity was measured as the concentration of adenosine triphosphate (ATP) within the community, measured using a Biotek Synergy 2 multimode plate reader and the BacTitr-Glo Cell Viability assay (Promega). There was a linear relationship between concentration and luminescence ($R^2 = 0.998$), which we used to convert luminescence to nM ATP. We measured the breakdown of substrates labelled with 4-methylumbelliferone (MUB). Samples were amended with 40 μM of the substrates (100 μl total volume) and incubated in the dark under the same conditions as the microcosms (static, 22 °C) for 60 min. After the incubation, 10 μl of 1 M NaOH was added and the

fluorescence was measured over 4 min with the maximum value recorded. Fluorescent values were converted to nM MUB after establishing a linear relationship between MUB concentration and fluorescence ($R^2 = 0.996$) and using negative controls to account for any autofluorescence in the medium. We selected substrates that were ecologically relevant to this ecosystem including xylosidase (cleaves the labile substrate xylose, a monomer prevalent in hemicellulose), $\beta$-chitinase (breaks down chitin, the main component of arthropod exoskeletons and fungal cell walls), $\beta$-glucosidase (breaks down cellulose, the structural component of plants) and phosphatase (breaks down organic monoesters for the mineralisation and acquisition of phosphorus).

## Sequencing methods

We characterised the composition of each initial community and of the four replicate final communities on the Illumina MiSeq platform by Molecular Research DNA. The V4 region of the 16S ribosomal RNA gene was amplified, using primers 515f (GTGCCAGCMGCCGCGGTAA)/806r (GGACTACHVGGGTWTCTAAT) with a barcoded forward primer. The sequencing effort (15000 reads per sample) was similar to the number of cells used to initiate the microcosms (~20,000 cells), so we assumed the communities were almost fully characterised. 16S rRNA amplicon sequence data were processed using the bash scripts and R (v4.2.1) programming language[32]. Briefly, demultiplexed sequence files obtained from the sequencing facility were processed using the DADA2 pipeline in R[33,34] version 1.14 to produce a bacterial amplicon sequence variant (ASVs: ref. 35) abundance table. The quality profiles of the reads were filtered and trimmed (using the function dada2::filterAndTrim); truncating sequences to 240bp (option trunLen = 240), removing reads with a quality score less than 11 (option truncQ = 11), discarding reads with ambiguous bases (Ns; option maxN = 0) or with more than one expected error (Ns; maxEE option = 1), and removing reads that matched the phiX genome (option rm.phix = TRUE). Error rates were then learned using dada2::learnErrors, before sample inference (ASV inference) using the main dada2::dada function to create an ASV abundance table. Chimeras were removed via the dada2::removeBimeraDenovo function using the consensus method (option method = 'consensus') before taxonomic assignment of the ASVs to the species level using dada2::assignTaxonomy, aligning to the SILVA v138 SSU Ref NR 99 database[36–38]. This process identified 21083 ASVs across all samples. After inference of ASVs from the sequence data in this way, we applied additional, custom quality-filtering procedures—namely, removing ASVs with fewer than 100 reads across samples (reducing the number of ASVs to 5834) and removing samples with fewer than 10,000 sequences (reducing the number of ASVs to 1209). This resulted in the final ASV table of 1209 ASV abundances across all of the samples from days 0 and 7.

## Statistics and reproducibility

Statistical methods and software used are described in the following sections and in figure legends.

## Determination of community classes

Following previous work, we determined community classes by computing all-against-all Jensen–Shannon divergence ($D_{JSD}$)[25] and performing a Partition Around Medoids clustering, which requires as input the number of output communities $k$. To find the optimal clustering, we ran the method for a broad range of $k$ values and computed the Calinski–Harabasz index ($CH$) that quantifies the quality of the classification. The optimal classification results from choosing $k_{opt} = \arg\max_k(CH)$, shown in Supplementary Fig. 4. This procedure was followed for each subset of data independently (after quality-filtering sequencing procedures, 658 samples for starting communities and 275 samples for each replicate of final communities).

## Optimal community superposition

Given two sets of paired points in the multidimensional space which, in this work, are determined by the relative abundances of starting communities and of one replicate of final communities, respectively, we asked if there was a rigid-body operation transforming the relative abundances of starting communities into the final ones. Such transformation, termed superposition, is a translation of the centroids of both sets to a common origin followed by a rotation minimising the Root Mean Square Deviation between both datasets. We searched for the optimal transformation using the Kabsch algorithm (adapting the implementation available in the URL: https://github.com/Fraternalilab/PDBencode), which applies the SVD of the cross-covariance matrix of the relative abundances of both datasets and then seeks an optimal transformation. To obtain an estimation of the quality of the superposition, we repeated the computation after considering as an input the starting matrix with the values of its cells randomly shuffled and the observed final matrix without modifications. We considered 50 randomisations to obtain a confidence interval (CI) of the RMSD. Finally, we further evaluated the prediction by representing the first and second components of the SVD of the transformed starting communities against those of the three remaining replicates of the final communities.

## Visual representation of the communities

ASV tables were rarefied to 10K reads for visualisation purposes only. Barplot representations were created with the R package PHYLOSEQ[39]. When classes were represented, communities within the same class were merged using the function MERGE_SAMPLES provided by phyloseq. Dimensionality reduction was performed by computing all-against-all communities dissimilarities with Jensen–Shannon divergence ($D_{JSD}$)[25], followed by a Principal Coordinates Analysis (R function DUDI.PCO, package ADE4[40]).

## Community classes and the compositional landscape

To evaluate the significance of the community classes that we identified, and to confront our categorisation with other potential groupings, we calculated the ANOSIM metric (R package VEGAN[41]) assessing the significance with permutation tests ($10^3$ permutations). To illustrate the fate of starting communities with respect to the final classes we first computed the centroid of each final class. For each set of communities belonging to a given class, we generated a new set by sampling 10,000 reads with the probabilities given by the relative abundances of each sample. The centroid was defined as the median value of each ASV in the resampled set, whose relative abundance was considered for downstream computations. Second, we computed, for each starting community, $D_{JSD}$ against all final communities of one of the replicates and its centroids. To generate Fig. 3D we identified the minimum distance to each class. Similar results were obtained regardless of the replicate that was selected.

## Estimation of ASVs propensities for subsets of communities

We defined a trajectory as a set of five communities that includes a parent community (a starting community) and the four replicates obtained from each parent (final community), see Fig. 6. Then, if the four final replicates converged to an equivalent final class we said that the trajectory was convergent, being divergent otherwise. These definitions split the ASV tables from the starting and final experiments into three types $T$ of communities: convergent to Class 1 (C1), convergent to Class 2 (C2) and divergent (D).

Next, let us consider that an ASV $i$ is observed in a community $a$, that is, $X_{ia}(t) > 0$ with $X_{ia}(t)$ being an ASV table at time $t = \{St, End\}$ (starting or final, the latter containing the four replicates). We computed the propensity that an ASV $i$ is observed in a generic community $a$ at time $t$, conditional on the community being classified into one of the three types of trajectories specified above (i.e. $\forall a, \exists! T \in \{C1, C2, D\}$

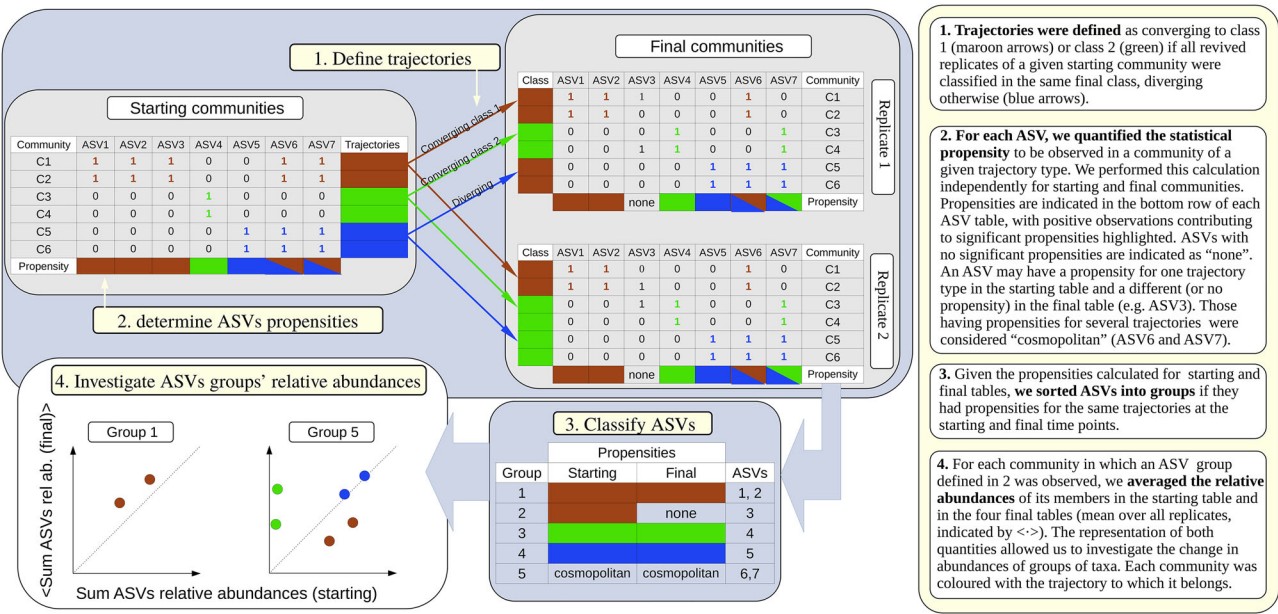

**Fig. 6 | Estimation of ASVs propensities.** The trajectories of the starting communities were identified and classified. This allowed us to calculate 'propensities': a table identifying whether ASVs were more likely observed in particular starting and final classes.

such that $a \in T$) as:

$$\text{Prop}(X_{ia}(t) > 0, a \in T) = \log\left(\frac{P(X_{ia}(t) > 0 \mid a \in T)}{P(X_{ia}(t) > 0)}\right). \quad (1)$$

Therefore, for each ASV we calculated six propensities (whether communities were associated with the three types of trajectories across the two time points). We estimated CIs by bootstrapping the ASVs table from which the correspondent propensity was calculated (i.e. repeatedly resampling communities with replacement). One thousand bootstrapped ASV tables were created for each time point and propensities were computed for each, then requiring that the 95% CI obtained for each propensity was positive. In Fig. 4 we studied changes throughout the two time points in the sum of the relative abundances of groups of ASVs having significant propensities for the same type of trajectories in the same time points.

**Metagenomic predictions**

Metagenomic predictions were performed using PiCRUST v2.4.2[42], which was also used to compute quality controls. Eleven ASVs had poor alignments and were removed from downstream analysis. The nsti score assessing the quality of the prediction was 0.042, indicating that the predictions were of high quality. Predictions were aggregated considering both the KEGG pathway hierarchy and BRITE annotations[27]. For each community class, the proportion of each pathway was averaged across all samples in that class. The difference in the mean proportions was then computed between each pair of classes, and those pathways showing a significant difference in at least one comparison were retained (Welch test corrected for multiple testing, the difference in mean proportions larger than 0.05 and corrected $p$ value < 0.001, see Supplementary Figs. 9 and 10 for examples). The mean proportion of the pathways selected was represented in a heatmap (Fig. 5a) and rescaled by computing a Z-score to highlight those over- (or under-) represented in each class. Rows and columns were clustered with an average linkage agglomerative clustering using an Euclidean distance (default method in ʜᴇᴀᴛᴍᴀᴘ.2, R package ɢᴘʟᴏᴛꜱ[43]).

**Reporting summary**

Further information on research design is available in the Nature Portfolio Reporting Summary linked to this article.

## Data availability

Sequences associated with this study are deposited at NCBI under BioProject accession number PRJNA989519. This project contains the 16S rRNA amplicon sequencing data associated with each of the communities at day 0, as well as at day 7 for the four replicate growth experiments. Source data and additional processed data are also available in the Code repository with details to reproduce data and figures (see "Code availability"). Source data are provided with this paper.

## Code availability

Code used for all the analysis presented in the manuscript was deposited in GitHub with the URL: https://github.com/apascualgarcia/ReplayEcology, and the first release was permanently stored in DOI: 10.5281/zenodo.13785758.

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

## Acknowledgements

We thank Lara Durán-Trío and Josep Ramoneda for useful discussions. The project was supported by an ERC Starting Grant to T.B. A.P.G. was funded by a Ramón y Cajal Fellowship from the Spanish Ministry of Science and Innovation (RyC2021-032424-I), by grant PID2022-139900NA-I00 (AEI/10.13039/501100011033/ FEDER, UE) and by the Simons Collaboration: Principles of Microbial Ecosystems, award 542381/FY22.

## Author contributions

Conceptualisation: T.B. Methodology: A.P.G., D.R., M.L.J., T.B. Investigation: A.P.G., D.R., M.L.J. Visualisation: A.P.G., T.B. Funding acquisition: A.P.G., T.B. Project administration: T.B. Supervision: T.B. Writing—original draft: A.P.G. Writing—review and editing: A.P.G., D.R., M.L.J., T.B.

## Competing interests

The authors declare no competing interests.
