## [Transparent Peer Review file · Nature Communications]

Replicating community dynamics reveals reveals how initial composition shapes the functional outcomes of bacterial communities

Corresponding Author: Professor Thomas Bell

Version 0:

Reviewer comments:

Reviewer #1

(Remarks to the Author)

The manuscript presents an exciting experimental study where the authors aimed to answer questions regarding the reproducibility and predictability of aquatic bacterial communities from a very special type of waterbody. It is definitely noteworthy to experimentally test over a thousand communities but I think the authors made a bit too far-reaching conclusions and wanted to generalise a bit too much to "sell" the MS. It was very useful knowledge to show that whenever one creates a frozen archive of communities and revive them, the community composition can be estimated to a great extent. But it remained unclear whether the communities were revived always after a certain time period or randomly within a longer timescale.

The "tipping-points" phenomenon was not thoroughly investigated, I believe. Here, I think there might be, and let me to speculate a bit, certain key taxa that potentially cannot be revived every time the authors tried. But if so, these key taxa might co-select several other taxa that eventually will result in a "more predictable" final class. I was missing some complementary analyses about this and discussion points about ecological co-selection or priority effects, in general. The environment that these bacteria originate are extremely dynamic ones and exposed to so many selective pressures. In the microcosms, the authors basically created the opposite of what these microbes are adapted to. For this reason, I missed some discussion about the potential effects of the lacking environmental fluctuations, moreover, whether the detected final community compositions (and classes) really represent permanent outcomes, or they are "lucky" snapshots that suggested predictability. I wonder what would have happened if the authors let them a few days more, or maybe if they revive them a year later.

The manuscript is well-written, however, I found it somewhat over-mystified and exaggerated. Just because the MS was written specifically for a Nature journal, it does not mean that the authors must use words that should fit better in a novel. There are also some parts that should be clarified. For example, it is unclear from the abstract what environment the authors have sampled and that it is about aquatic bacterial communities. Try to avoid odd wordings as specified below in my comments. And when I saw Fig 1, I did not even expect that the sampled communities originate from aquatic environments. Having the Methods in the end requires the authors to introduce as many essential information as possible in the Introduction and Results.

Figure 4 has a very catchy illustration but it does not align completely with the observed patterns. The term attractor is also catchy but what, or rather, who is behind it? How stable is the "stable" really?

Taken together, this study presents an extremely great and hard work (congratulation!), but it gives little novelty and robustness that would be necessary for a Nature journal. The authors really tried to generalise every single result, which I found a bit problematic in this case.

Specific comments:

L27: of course, think about priority effects, for example

L19: "complex environment" - too vague phrasing

L28-30: too general and lacks novelty

L67-68: Compositional or functional?

L69: "complex communities" - way too vague phrasing. Clarify what it is!

L78: "wild" - weird wording, simply use 'aquatic'

L79: wild -> let's call it 'source'
L81: Unclear what time-window occurred between preservation and when they got revived.
L89: Are these the cryopreserved and revived or the initial communities?
L92: So no stochastic assembly?
L93: relative abundance?
L95: direction of travel -> shift; again, this should not be written like a novel
L104-106: Very great clear sentence!
L106: Could it be the presence of ecological co-selection?
L109: Still wondering how much time the communities spent frozen.
L113: insert (classes) before the comma
L124: twice? Not four times?!
L126: Fig. 2 -> Fig. 2b
L132-134: Nice and important sentence! Although it is not so valid for classes 3 and 5.
L138: reference for KEGG database?
L140: Majority? Half.; were -> was
L149: "fewer": really? It is not visible clearly on the figure.
L150-151: "leaf litter uptake" - very odd phrasing, plus, these ephemeral waterbodies are surrounded by the tree bark and there are numerous additional ecological processes than leaf litter decomposition.
L153: "degradation rate" - rate should be used only if you have a time unit in the measured values
L163: If not neutral community drift (stochasticity), then what drives this divergence when the environmental conditions are the same?
L168: maybe: of the five initial starting community. Also, numbers below 10 should be written with letters.
L172-173: Reference is missing.
L175-177: Thus, diminishing the predictability of trajectories.
L186-203: I found it a bit too much and unnecessary description.
L202: remove the word 'community'
L203-304: Or not, since some classes tend to have different trajectories in ~50% of the cases.

Figure 1: (a) part should be labelled with starting and final communities. The tree icons should be replaced with drawings that are more accurately illustrate the aquatic environment. "Sequenced + Function measurements" -> maybe "Estimation of community composition and functions". (b) I think it can be removed from here. I would rather see a boxplot that shows the community distance between starting and final communities (each data point is a distance between its starting and final community). In the caption: "community composition measured" - it should be assessed not measured. Number of sampled sites should be stated. The explanation of the method for the (c) part is difficult to understand without reading the Methods. "Left panel", what do you mean?

Figure 2: It took some time before I realised that classes are not taxonomic classes here. What about if you would call/label them as clusters instead? Caption: "final community classes", based on unsupervised clustering... needs to be specified. "the proportion" is correct but the percentage symbol should be removed from the legend.

Figure 3: Use the same colours as in Fig. 2a.

Figure 4: Why not use the same colour as the (b) part? Or the ones that you used in Fig. 2a? Similarly, I would use the numbers instead of letters for each classes.

Reviewer #2

(Remarks to the Author)

In this manuscript the authors demonstrate that 275 naturally occurring bacterial communities change over time in a largely reproducible manner. Their analyses stem from the dimensionality-reduction techniques of principal coordinate analysis and unsupervised clustering. The authors cluster the initial communities into 5 community classes and the final communities into 2 community classes, and based on this class decomposition postulate a landscape that could account for the observed microbial dynamics. The authors perform a metagenomic analysis to contrast functions of the two final community classes that seems reasonable, but which I am unable to critically evaluate.

Overall, this is an expertly conducted study that sheds light on the reproducibility and predictability of microbial ecosystems. A strength of the paper is that the authors manage to take an analysis of complex bacterial community dynamics and, through data analysis methods, make it intelligible to a broad readership.

However, this strength is also the cause of my principal concern regarding the paper, which is that it depends heavily on opaque and nonlinear dimensionality-reduction methods (that have been critiqued in e.g. Warton et al., *Methods in Ecology and Evolution* 2012). I recognize that given a community snapshot of 10,000 ASVs, these types of methods are natural if not inevitable. Nonetheless, since many of the results in this manuscript stem from PCoA, I recommend that the authors discuss the general limitations of PCoA and the considerations that led them to use it in this instance.

Furthermore:

+ Since the authors previously published a similar compositional class analysis (Pascual-Garcia and Bell, Nature Communications 2020), the authors would provide a service to the microbial ecology community by contrasting the inferred starting community classes in this manuscript with the inferred classes of their older work. To wit, how robust these community class inferences?

+ Explanations of data analysis procedures were often lacking and could be fleshed out. In particular:

+ Am I correct in assuming that the 2D PCoA spaces in Figures 1b and 2a calculated from the all-against-all dissimilarity matrix that takes into account all communities (i.e. both initial and final communities), and that Fig 1b only plots the starting communities on the left and the final communities on the right? If so, please clarify in the text.

+ In Figure 1 are you performing a "translation and rotation" in high-dimensional ASV space or in ordination space? I found the explanation of this SVD analysis unclear.

+ How are classes represented as a bar plot in Figure 2b? A given class consists of many communities-- is the bar plot plotting the mean relative abundance for communities of a given class, or using the centroid of a given class (as described on line 319), or something else?

+ Figure 4c: It is not clear what "mean distance to borders" means.

Lastly, a few minor typos/comments:

+ line 100: why are both bounds of the 95% CI greater than the mean?

+ line 143: missing period

+ line 139 and on: inconsistent capitalization of "class"?

+ Fig 4a: "stable" and "less stable" seem to me like descriptions of the basin, not of the z-axis. Perhaps instead, a downward arrow labeled by "community dynamics", or no z-axis label at all?

+ Fig 4 caption: Missing period before (C)

+ line 242: independently

+ line 305: not clear what "superposition" means

+ line 332: p-val

+ What does "Parent" mean in the SI?

Version 1:

Reviewer comments:

Reviewer #1

(Remarks to the Author)

Dear Authors,

Many thanks for addressing all my comments and responding to my concerns! I believe that the clarifications throughout the text and the new figures made the manuscript acceptable for publication as it now nicely conveys the results with clear interpretations and conclusions.

Reviewer #2

(Remarks to the Author)

My main concern is that large parts of the newly added text and figures are very confusing to me, to the point that I cannot understand what the authors mean after several attempts. Unfortunately, in my opinion the manuscript has not been improved following revision.

The subsection "Core sets of ASVs determined convergence towards the attractors" relies on the "propensity", whose mathematical definition is something like the log-fold change in the probability of observing an ASV in a "convergent to Class 1" trajectory compared to the probability of observing that ASV in any trajectory. This definition is crucial for all of the subsequent analysis in the subsection and Figures 5 and 6, but its mathematical definition and graphical description (Figure 5a) are very difficult to parse, and I do not believe a typical reader of Nature Communications would understand what it means. The remaining content in this subsection is overly detailed (and more likely should belong in the supplement than the main text) and fails to convincingly summarize their later claim of "identifying two tipping points" that determine the compositional outcome of the replicates. Figure 6 is not even mentioned in the main text.

I have a few other minor points that I hope the authors find useful:

+ line 114: The sentence mentioning the "small and significant Root Mean Square Deviation" does not communicate the result that the rigid-body transformation works better with the real data than for a null model in which the starting communities are shuffled, which I think is what the authors mean to say

+ line 130: should be "communities'" (with an apostrophe)

+ line 144: while OTU vs ASV certainly introduces some discrepancy, unsupervised clustering must also play some role

+ line 146: grammatically, should be "providing strong evidence of selection"

+ line 452: How did the authors "bootstrap the data 1000 times"? There are many ways to bootstrap an ASV table.

+ line 453: The propensity can be negative, yet the authors specify that they "require" the 95% CI to be positive. Why?
+ I am still not sure how to interpret Figure 4c, and I also find the final sentence of the caption of Figure 4 confusing, since it seems that the green and brown points in Fig 4c are distinguished by the minimum distance to centroids, and are not strongly affected by the mean distance to borders.

Version 2:

Reviewer comments:

Reviewer #2

(Remarks to the Author)

I am satisfied with the authors' clarifications.

REVIEWER COMMENTS AND AUTHOR RESPONSE

Reviewer #1 (Remarks to the Author):

The manuscript presents an exciting experimental study where the authors aimed to answer questions regarding the reproducibility and predictability of aquatic bacterial communities from a very special type of waterbody. It is definitely noteworthy to experimentally test over a thousand communities but I think the authors made a bit too far-reaching conclusions and wanted to generalise a bit too much to "sell" the MS. It was very useful knowledge to show that whenever one creates a frozen archive of communities and revive them, the community composition can be estimated to a great extent. But it remained unclear whether the communities were revived always after a certain time period or randomly within a longer timescale.

AUTHORS: We thank the reviewer for sharing their appreciation and excitement for our work. We took note about the "overselling" comment whenever specific points were raised by the reviewer and have changed the language accordingly.

For the time period: Briefly, the communities were not revived randomly, but all communities were revived together at the same time. The communities were collected and cryopreserved over a period of around a year as detailed in other publications (Pascual-García and Bell, *Nature Commun.* 2020). Once cryopreserved at -80C, the communities were static and we would not anticipate any degradation of the sample (beyond the impact of cryopreservation and thawing, which we have detailed elsewhere (Rivett and Bell, *Nature Microbiol* 2018)). To ensure that there was no systematic bias in our experimental procedure, we thawed all communities at the same time for the experiments. The communities were thawed in this way on two separate occasions to conduct four independent trials of the experiment. We clarify these points in the Methodology.

The "tipping-points" phenomenon was not thoroughly investigated, I believe. Here, I think there might be, and let me to speculate a bit, certain key taxa that potentially cannot be revived every time the authors tried. But if so, these key taxa might co-select several other taxa that eventually will result in a "more predictable" final class. I was missing some complementary analyses about this and discussion points about ecological co-selection or priority effects, in general.

AUTHORS: We thank the reviewer for this comment, which motivated us to explore our results in more detail, focusing on key taxa rather than communities as a whole. We hope these extensive new analyses provide the ecological insights that the reviewer was suggesting. In particular, we have added analyses motivated by this comment in which we studied how the presence/absence and relative abundances of the different taxa changed between the starting and final experiments. This analysis included species that were not detected. Please note that we refer to "undetected species" and rather than "non-revived" species because there may be species present at the starting point that may have been revived but did not grow sufficiently to be detected.

The analysis was conducted in two steps.

Step 1

- We defined a “trajectory” as a set of five communities containing a given starting community and the final community for its four revived replicates.
- Each trajectory was then classified into one of the following 3 types:
 1. Trajectories converging to Class 1 (all the 4 revived replicates were clustered in Final Class 1),
 2. Trajectories converging to Class 2 (all 4 replicates clustered in Final Class2)
 3. Diverging trajectories (replicates were not consistently clustered in the same final class).
- The classification of the trajectories into these 3 types allowed us to split communities in the ASVs tables (both starting and final) into these 3 types.
- Next we computed the statistical *propensity* for an ASV to be observed in a community, given that the community belongs to one of the three types of trajectories. The propensity was computed for the starting and final ASV tables independently (therefore 6 propensities: 2 time points x 3 trajectories). A formal definition of the propensity is given in the Methods.

Step 2

- We identified groups of ASVs that had similar statistical propensities. For example, one group could be the set of ASVs having a significant statistical propensity for communities in trajectories converging to Class 1 at the starting time point, and for communities in divergent trajectories in the final time point.
- We studied changes in the relative abundances of the different ASVs groups. This allowed us to observe which ASVs groups had the greatest changes between starting and final communities and how that depended on the trajectories to which communities belonged.

Our study revealed an interesting group of ASVs having a significant propensity (both at the starting and final time points) for communities with trajectories converging to Class 1, hereafter **S&F1** group.

We further observed that, at the starting point, the sum of S&F1 relative abundances was typically higher than 0.12 for communities belonging to Class 1 trajectories, while it was lower than 0.12 for those belonging to Class 2 trajectories. In the few cases ASV abundances were higher than 0.12 for Class 2 communities at the starting point, it became reduced below this level by the end point. The high abundance of S&F1 ASVs seem to determine convergence to the Class 1 attractor, which is consistent with the pattern that would be observed if there were strong priority effects.

We also observed a second group of ASVs that were significantly present in all trajectories, which we termed “cosmopolitan” ASVs. Cosmopolitan ASVs had a contrasting pattern to S&F1 ASVs: their relative abundances were high in Class 2 trajectories, and they were low in Class 1 trajectories. It was also possible to determine a threshold determining convergence to the Class 2 attractor.

Finally, combining both S&F1 and cosmopolitan groups observations, we found that a necessary condition to observe divergent trajectories was having a balance in the relative abundances of both groups.

There are related observations, explained in the manuscript, which imply “co-dependencies” among ASVs.

We note that, without explicit manipulation of these groups within the communities we cannot confirm these ecological mechanisms, but they do provide illuminating patterns against which future studies could be compared.

As suggested by the reviewer, we also explored whether non-detected ASVs influenced the trajectories. Non-detected ASVs were defined as those present in the starting communities but not in final communities. Non-detected ASVs had a similar quantile distribution as the other ASVs, except the highest quantile (95%) was around two orders of magnitude smaller than the resurrected ASVs. We therefore excluded the possibility that the most abundant ASVs were not resurrected. In addition, non-detected ASVs contributed only 3.45% of the sequencing reads in the starting communities. Those that had some statistically significant propensity were associated with Class 1 (the dominant attractor), and they were present at a mean relative abundance <0.004 in communities converging to Class 2 or diverging. Therefore, it is unlikely they influenced the formation of the final attractors or determined the trajectories. We cannot rule out that they had an influence on the appearance of ASVs observed at very low abundances at the start of the experiment since they had a non-negligible abundance and an affiliation to Class 1 at the end of the experiment.

In summary, the trajectories seemed to be explained by groups of taxa present in both starting and final communities at high abundances, and not by individual “keystone” species whose presence/absence may have had dramatic effects on the dynamics even at low abundance.

The requested analyses (and the conclusions drawn from them) are fairly substantial and complex, but we felt they would be of interest to a general readership interested in how to gain insight from community trajectories, so we have tried to summarise the key findings in the text.

The environment that these bacteria originate are extremely dynamic ones and exposed to so many selective pressures. In the microcosms, the authors basically created the opposite of what these microbes are adapted to. For this reason, I missed some discussion about the potential effects of the lacking environmental fluctuations, moreover, whether the detected final community compositions (and classes) really represent permanent outcomes, or they are "lucky" snapshots that suggested predictability. I wonder what would have happened if the authors let them a few days more, or maybe if they revive them a year later.

AUTHORS: In previous work (Pascual-García and Bell, *Nature Commun*, 2020), we showed that, after inoculating the communities into this growth medium, it was possible to find a strong distance-decay relationship (communities that were collected from closer locations had more similar community composition) spanning several orders of magnitude in space. Indeed, clustering communities according to the specific location from which they were sampled provided a statistically significant classification of the communities. This means that, although communities were grown in the beech medium under standardised and controlled conditions of the lab, we were able to capture their history and provenance. The observation that these characteristics were retained despite regrowth in the lab conditions provides a strong justification for the laboratory system.

Since this observation (strong distance decay in community similarity) could simply be a consequence of dispersal limitation, we further performed an unsupervised clustering of these communities (Pascual-García and Bell, *Nature Commun*, 2020). Interestingly, we found a classification that was even more significant than the one found by clustering them according to their locations, comprising only six classes. Importantly, communities distant in space often clustered together, and we found distinct metagenomic and functional signatures describing these six clusters that could be interpreted in terms of selection, ruling out stochastic effects as an explanation. Therefore, we have good reasons to think that these are permanent outcomes and not “lucky snapshots”. Indeed, subsequent work (Rivett et al *ISME J* 2021) suggested that the size of the tree-hole is the main variable determining the composition of these communities. Therefore, the medium selected seems appropriate to study these communities, because it allows us to detect both history and signatures of environmental filtering. We added some more context in the section describing the classes to clarify these points.

The manuscript is well-written, however, I found it somewhat over-mystified and exaggerated. Just because the MS was written specifically for a *Nature* journal, it does not mean that the authors must use words that should fit better in a novel.

AUTHORS: We have carefully read through the manuscript to try to identify any exaggerations. We have tried to address any specific comments that the reviewer has provided.

There are also some parts that should be clarified. For example, it is unclear from the abstract what environment the authors have sampled and that it is about aquatic bacterial communities. Try to avoid odd wordings as specified below in my comments. And when I saw Fig 1, I did not even expect that the sampled communities originate from aquatic environments. Having the Methods in the end requires the authors to introduce as many essential information as possible in the Introduction and Results.

AUTHORS: Samples were collected from the rainwater pools that collect in the buttressing of beech tree holes. These aquatic habitats are often permanently wet but can dry out periodically which we suspect is why some taxa are also found in soil communities. We clarified these points in Introduction and Results and modified Fig. 1 to illustrate more clearly the environment.

Figure 4 has a very catchy illustration but it does not align completely with the observed patterns. The term attractor is also catchy but what, or rather, who is behind it? How stable is the "stable" really?

AUTHORS: The term ‘attractor’ is a concept from the mathematics of dynamical systems and is very widely used in theoretical ecology for describing population and community dynamics. An attractor is ‘a set of states toward which a system tends to evolve for a wide variety of starting conditions of the system’ (from Wikipedia: en.wikipedia.org/wiki/Attractor). In our study, we use the term attractor both as a phenomenological observation and a hypothesis. The observation is that the compositions of the different communities move towards a specific region in the compositional landscape. The hypothesis is that this happens because these compositions represent the best adapted combinations of species and abundances for the environmental conditions in the microcosms.

We have used this observation and hypothesis to probe potential underlying mechanisms- for example, the communities may shape their own environment through consumption of resources and cross-

feeding interactions. The consequence is that the number of available nutrients would be depleted to a minimum, fostering stability and non-invasibility. The z-axis of the figure could represent “community stability”, “community-level fitness” or, in the opposite direction, “nutrient availability”.

The figure is meant to capture the overall idea rather than the specifics of our results, which are covered in the figures that follow from this conceptual diagram.

Taken together, this study presents an extremely great and hard work (congratulation!), but it gives little novelty and robustness that would be necessary for a Nature journal. The authors really tried to generalise every single result, which I found a bit problematic in this case.

AUTHORS: We agree with the reviewer that, to merit publication in a Nature journal, the contribution needs to be substantial, and we are grateful to the reviewer for acknowledging the substantial body of work presented here.

The reviewer raises the question of novelty, which we agree must also be high for this calibre of journal. Unfortunately, the reviewer did not provide any other peer-reviewed publication against which to compare our results, making it difficult to respond to their comment other than to reiterate that we are unaware of a comparable study in the microbial literature or even more broadly in the ecological literature.

We acknowledge here and in the manuscript that there is a mature literature on community attractors and multiple stable states, but we are unaware of any study that has attempted what we have done here: to initiate the development of hundreds of communities in a standardised environment. To the best of our knowledge, this has not been attempted previously in the context of looking at community trajectories, to date making it the most comprehensive (only?) study of its kind using complex communities and a complex environment.

The demonstrated predictability in the community trajectories is an inescapable prerequisite for the domestication of complex communities, which is still far from being widely explored in the field- we hope the study will be a reference point for the future development of a top-down synthetic ecology. While we have avoided making big claims about domestication in the manuscript, it is clear that there are some novel applications that follow from this line of research.

Although the generalisability remains to be seen, we feel there is a reasonable expectation that the results are generalisable to microbial communities because we have assayed hundreds of communities. Smaller scale experiments using fewer communities could not make the general claims we make here.

Specific comments:

L27: of course, think about priority effects, for example

AUTHORS: As described in the Methods, the communities were grown to stationary phase before they were cryo-preserved. The different compositional classes had a strong signal of selection and so any priority effects would already have happened prior to the start of the experiment. When the

communities were revived, priority effects could have happened if there were keystone species that were/were not revived, which would have strongly conditioned the trajectories. Our new results have discarded this possibility. Since there was no immigration, we could not see how priority effects could otherwise be considered.

L19: "complex environment" - too vague phrasing

AUTHORS: We rephrased it to "complex resource environment" to distinguish it from studies that use minimal media amended with a defined and small number of resources.

L28-30: too general and lacks novelty

AUTHORS: We respectfully disagree- we discuss novelty in the response to one of the general comments above.

L67-68: Compositional or functional?

AUTHORS: We rephrased it as: "does replaying the tape of ecology produce the same compositional and functional outcome?"

L69: "complex communities" - way too vague phrasing. Clarify what it is!

AUTHORS: We included: "(i.e. hosting hundreds of different taxa)".

L78: "wild" - weird wording, simply use 'aquatic'.

AUTHORS: We used "wild" to emphasize that they are not 'synthetic' communities using 'domestic' lab strains. We now use "natural" communities.

L79: wild -> let's call it 'source'

AUTHORS: Thanks, we modified it accordingly.

L81: Unclear what time-window occurred between preservation and when they got revived.

AUTHORS: The communities were cryo-preserved at -80C. At such low temperatures there is no bacterial activity, so the time window between preservation and revival is not relevant since the communities are in stasis. However, to answer the question for the information of the reviewer- the communities were originally collected from August 2013 to April 2014 and the experiments were initiated in autumn 2014. Details of the collection and initiation of the experiment are published elsewhere [Rivett and Bell, Nature Microb 2018, Pascual-García and Bell, Nature Commun. 2020]. In the Rivett and Bell (2018) article, we also provided sequencing information on the impact of cryopreservation on the communities.

L89: Are these the cryopreserved and revived or the initial communities?

AUTHORS: There is only one replicate for the starting (cryopreserved) communities and four replicates for the end (revived) communities. We modified (in bold) the phrase as:

“We first identified whether the 4 revived replicates of each of the 275 cryopreserved communities were clustered at the end of the experiment”

L92: So no stochastic assembly?

AUTHORS: We added “hence ruling out stochastic assembly”

L93: relative abundance?

AUTHORS: Thanks for pointing out, corrected.

L95: direction of travel -> shift; again, this should not be written like a novel

AUTHORS: We are grateful to the reviewer for their careful reading of the wording of the ms. We have changed to “direction of the shift” since it is the direction that is important, not just that they shifted.

L104-106: Very great clear sentence!

AUTHORS: Thank you.

L106: Could it be the presence of ecological co-selection?

AUTHORS: In the new results we addressed this question (see lengthy response above), and it seems to be the case that there is a group of taxa dominating communities with trajectories towards class 1 that co-select another group of taxa. Perhaps more relevant, the absence of these key taxa appears to facilitate the presence of other taxa which, in turn, either diverge or converge to class 2. These new results do imply ecological co-selection and are now presented in a new section with a new figure to accompany these results

L109: Still wondering how much time the communities spent frozen.

AUTHORS: See comment above re. frozen communities. The communities were frozen for several months but we do not believe this is relevant and that we could conduct the same experiment with the same communities even after several years. We are still using the same communities today.

L113: insert (classes) before the comma

AUTHORS: Done.

L124: twice? Not four times?!

AUTHORS: Here we meant that communities were grown in the beech-tea medium before cryopreservation (once) and after being revived. i.e. each community was therefore regrown twice). We rephrased it as:

“The communities retained signatures of their provenance despite being re-grown twice on the beech leaf media (they were first grown to stationary phase before cryopreservation and then second revived and grown across four replicates).”

L126: Fig. 2 -> Fig. 2b

AUTHORS: Corrected.

L132-134: Nice and important sentence! Although it is not so valid for classes 3 and 5.

AUTHORS: There are two results to take into account here. First, whether, given one revived community, its four propagated replicates converged to the same class (the result this phrase is referring to). Second, whether communities revived from the same starting class end up in the same final class (the result the reviewer refers to). To make more clear this distinction we added:

“However, the specific class to which revived communities converged depended on the starting class: starting classes 1, 2 and 4 tended to end up in the same final class, while starting classes 3 and 5 were more unpredictable”.

L138: reference for KEGG database?

AUTHORS: Thank you, we included it now.

L140: Majority? Half.; were -> was

AUTHORS: We indicated “more than two-thirds” and refer to Suppl. Table 1 where specific values are reported.

L149: "fewer": really? It is not visible clearly on the figure.

AUTHORS: We now say “less than one-third” and refer to Suppl. Table 1.

L150-151: "leaf litter uptake" - very odd phrasing, plus, these ephemeral waterbodies are surrounded by the tree bark and there are numerous additional ecological processes than leaf litter decomposition.

AUTHORS: We changed “litter” to “medium”.

L153: "degradation rate" - rate should be used only if you have a time unit in the measured values

AUTHORS: The exoenzymatic activities consider the amount of fluorescent moiety cleaved from the substrate backbone in 60 minutes of incubation.

L163: If not neutral community drift (stochasticity), then what drives this divergence when the environmental conditions are the same?

AUTHORS: The starting composition of the revived communities.

L168: maybe: of the five initial starting community. Also, numbers below 10 should be written with letters.

AUTHORS: Corrected

L172-173: Reference is missing.

AUTHORS: We included the reference. Thank you.

L175-177: Thus, diminishing the predictability of trajectories.

AUTHORS: Thank you, added.

L186-203: I found it a bit too much and unnecessary description.

AUTHORS: We reduced the description.

L202: remove the word 'community'

AUTHORS: Done

L203-304: Or not, since some classes tend to have different trajectories in ~50% of the cases.

AUTHORS: In these lines (assumed 203-204) we wrote: "As illustrated here, rugged landscapes may exhibit convergent or divergent outcomes depending on the starting location in compositional space."

We cannot see how the reviewer's comment connects to this sentence, but we feel it could be connected with our previous answer, namely that there is a distinction between reproducibility (given a revived community, whether the different replicates follow the same trajectory, i.e. the "tape") and predictability (given a set of communities, whether their replicates will follow similar trajectories). Both are of course influenced by the landscape, but it is in the latter case when it becomes apparent.

Figure 1: (a) part should be labelled with starting and final communities. The tree icons should be replaced with drawings that are more accurately illustrate the aquatic environment. "Sequenced + Function measurements" -> maybe "Estimation of community composition and functions". (b) I think it can be removed from here. I would rather see a boxplot that shows the community distance between starting and final communities (each data point is a distance between its starting and final community). In the caption: "community composition measured" - it should be assessed not measured. Number of

sampled sites should be stated. The explanation of the method for the (c) part is difficult to understand without reading the Methods. "Left panel", what do you mean?

AUTHORS: We now include a picture of a water-filled tree hole -and incorporated all reviewer's comments except for Fig. 1b, which we keep in its current form since it shows how the ordination changed from starting to ending communities (encoded in the colour of the data points), which is an important point that we refer to in the manuscript. The boxplot suggested by the reviewer, showing the distance between starting and final communities, would not provide this important information about the direction of the trajectories.

Figure 2: It took some time before I realised that classes are not taxonomic classes here. What about if you would call/label them as clusters instead? Caption: "final community classes", based on unsupervised clustering... needs to be specified. "the proportion" is correct but the percentage symbol should be removed from the legend.

AUTHORS: The classes are based on the composition of the communities- they are clusters of communities in ordination space. To clarify, we incorporated the following: "of all communities with colours representing the starting and final community classes (compositionally-similar clusters of communities determined by unsupervised clustering)" used the term classes in previous work, so we would like to stick to this term for consistency. We corrected the caption. The (%) symbol is needed because the relative abundances are not presented in the interval [0, 1] but [0, 100].

Figure 3: Use the same colours as in Fig. 2a.

Figure 4: Why not use the same colour as the (b) part? Or the ones that you used in Fig. 2a? Similarly, I would use the numbers instead of letters for each classes.

AUTHORS: We modified the colours to make them consistent across the three figures. Please note that letters refer to communities, while classes are represented with colours.

Reviewer #2 (Remarks to the Author):

In this manuscript the authors demonstrate that 275 naturally occurring bacterial communities change over time in a largely reproducible manner. Their analyses stem from the dimensionality-reduction techniques of principal coordinate analysis and unsupervised clustering.

AUTHORS: Please note that principal coordinate analysis was used only for visualization. Classes were determined by unsupervised clustering of the relative abundances in the multidimensional space.

The authors cluster the initial communities into 5 community classes and the final communities into 2 community classes, and based on this class decomposition postulate a landscape that could account for the observed microbial dynamics. The authors perform a metagenomic analysis to contrast functions of the two final community classes that seems reasonable, but which I am unable to critically evaluate.

Overall, this is an expertly conducted study that sheds light on the reproducibility and predictability of microbial ecosystems. A strength of the paper is that the authors manage to take an analysis of complex bacterial community dynamics and, through data analysis methods, make it intelligible to a broad readership.

AUTHORS: Thanks for the summary and for the positive comments.

However, this strength is also the cause of my principal concern regarding the paper, which is that it depends heavily on opaque and nonlinear dimensionality-reduction methods (that have been critiqued in e.g. Warton et al., *Methods in Ecology and Evolution* 2012). I recognize that given a community snapshot of 10,000 ASVs, these types of methods are natural if not inevitable. Nonetheless, since many of the results in this manuscript stem from PCoA, I recommend that the authors discuss the general limitations of PCoA and the considerations that led them to use it in this instance.

AUTHORS: We acknowledge these caveats and this is the reason why the classification was based on unsupervised clustering of Jensen-Shannon distances considering the multidimensional relative abundances profiles, and not on PCoA, which was used for visualization only. We reviewed the text to make this point clear. We have also now added some extensive new analyses and visualisation that look at individual taxa- although these new analyses also make use of the community classes, they give a more intuitive idea of how taxonomic groups are reacting to their environment.

Furthermore:

+ Since the authors previously published a similar compositional class analysis (Pascual-Garcia and Bell, *Nature Communications* 2020), the authors would provide a service to the microbial ecology community by contrasting the inferred starting community classes in this manuscript with the inferred classes of their older work. To wit, how robust these community class inferences?

AUTHORS: Thanks for the suggestion. We added some text relating both works in the first section of Results. Unfortunately, we used two different bioinformatics pipelines for processing the sequencing since, at the time of the first publication, it was not customary to use ASVs. Therefore, the classification made at that time was based on OTUs. Despite these differences, both classifications are fairly robust. To compare both classifications we compared all-against-all communities asking whether each pair was classified in the same cluster in both classifications. The number of pairs classified in any cluster was the same in both classifications (~46K). Since the number of clusters was the same, this finding indicates that the size distribution of the clusters was similar. More importantly, ~50% of pairs of communities (~24K) were classified in the same cluster in both classifications.

+ Explanations of data analysis procedures were often lacking and could be fleshed out. In particular:
+ Am I correct in assuming that the 2D PCoA spaces in Figures 1b and 2a calculated from the all-against-all dissimilarity matrix that takes into account all communities (i.e. both initial and final communities), and that Fig 1b only plots the starting communities on the left and the final communities on the right? If so, please clarify in the text.

AUTHORS: Both figures were generated from an all-against-all dissimilarity matrix considering all data. As it can be seen from the scales in the x axis and for the amount of variance explained, both figures are the same, the difference being how we coloured the dots and split the data in boxes for visualization. In the new version, we explicitly indicate it in the caption of Fig 2.

+ In Figure 1 are you performing a "translation and rotation" in high-dimensional ASV space or in ordination space? I found the explanation of this SVD analysis unclear.

AUTHORS: The transformation was obtained in the high-dimensional ASV space considering the starting data and only one of the final replicates. We included more information in the caption and clarified the methods.

+ How are classes represented as a bar plot in Figure 2b? A given class consists of many communities-- is the bar plot plotting the mean relative abundance for communities of a given class, or using the centroid of a given class (as described on line 319), or something else?

AUTHORS: Samples were first rarefied, and then those belonging to the same class were merged using the function `merge_samples` in the R package *phyloseq*. This function simply sums the (rarefied) abundances of all samples belonging to the same level in the factor indicated (in this case, the class). Then, the relative abundances of the sum was used for subsequent visualisations. The outcome is an average community for each of the community classes. We added this explanation in Methods.

+ Figure 4c: It is not clear what "mean distance to borders" means.

AUTHORS: We computed the all-against-all distances between starting and final communities and then we considered, for each starting community, the nearest community associated with each attractor (i.e. two distance: distance to nearest Class 1 community and distance to nearest Class 2 community) as the "nearest border". We then took the arithmetic mean of both distances. In the plot, we considered only one of the replicates since similar results were obtained for the remaining replicates. We have extended the definition in Main Text and in the caption.

Lastly, a few minor typos/comments:

+ line 100: why are both bounds of the 95% CI greater than the mean?

AUTHORS: Thanks for pointing this out because we realized that our procedure was not clearly explained. There are several procedures one could use to investigate whether the RMSD obtained was significant. Since there are two matrices (starting community and one replicate of the final communities) that we aim to superimpose, when we find the optimal transformation we could quantify and compare the RMSD:

1. considering a starting random matrix as an input (generated by e.g. a given distribution) and the observed final matrix without modifications.
2. considering as an input the starting matrix with the values of its cells randomly shuffled and the observed final matrix without modifications.

3. considering as an input a bootstrapped starting matrix and the observed final matrix without modifications.
4. considering as an input bootstrapped starting and final matrices (bootstrapped pairs)

The first approach generates a much higher RMSD interval (>2), which allows us to know that the RMSD obtained was meaningful. The second and third provided intervals that were also higher than the observed value. For (3), the interval was higher than the mean because the final matrix was not bootstrapped, so it does not differ much from the second approach. Interestingly, (4) provides an interval lower than the observed value because a paired bootstrap favours finding an optimal transformation.

We felt (3) was the best approach, but, since the explanation was too convoluted to be easily followed, we now present (2), which we feel is easier to grasp and provides similar information and results.

+ line 143: missing period

AUTHORS: Corrected.

+ line 139 and on: inconsistent capitalization of "class"?

AUTHORS: Checked and remedied.

+ Fig 4a: "stable" and "less stable" seem to me like descriptions of the basin, not of the z-axis. Perhaps instead, a downward arrow labeled by "community dynamics", or no z-axis label at all?

AUTHORS: "More stable" and "less stable" are descriptions of the communities that are placed in this ordination space- the basins map those patterns of stability onto the ordination space. There are many definitions of stability used in the ecological literature, but we do not specify a particular definition because the idea would generally apply to a range of definitions. The surface shows how 'stable' a community would be if it were at a particular position on the x- and y- ordination axes. A community that is "more stable" (z axis) would tend to stay at that location in ordination space (or return to that location, depending on the stability definition), so it is indeed linked to the community dynamics. If 'community dynamics' was on the z-axis, it was unclear how that would be measured (e.g. what units?).

We agree with the reviewer that "stability" could refer to the width of the basin, although it could also refer to its height (as it may be the case for the mean rate of return, sometimes termed resilience, of a linearized system in local stability analysis).

We do feel we need some kind of z-axis label so that the figure is interpretable. We have added "Community stability" to the axis label to clarify that we are referring to the communities shown on the figure.

+ Fig 4 caption: Missing period before ©

AUTHORS: Corrected

+ line 242: independently

AUTHORS: Corrected

+ line 305: not clear what "superposition" means

AUTHORS: Given two paired sets of points, their superimposition (=superposition) is a rigid-body transformation which first performs a translation to make the centroids of both sets coincide followed by the search of a rotation matrix such that the RMSD between both sets is minimized. We modified the Methods to make this definition more explicit.

+ line 332: p-val

AUTHORS: Corrected

+ What does "Parent" mean in the SI?

AUTHORS: The term 'Parent' refers to each of the 275 starting communities and 'Children' to their four revived replicates. Therefore, 'Parent+Children' means each of the 275 groups comprising each parent and its children and 'Children' to the same classification excluding parents. This explanation was included in the new version.

Reviewer comments are in black. Author responses in red.

Reviewer #1 (Remarks to the Author):

Dear Authors,

Many thanks for addressing all my comments and responding to my concerns! I believe that the clarifications throughout the text and the new figures made the manuscript acceptable for publication as it now nicely conveys the results with clear interpretations and conclusions.

Reviewer #2 (Remarks to the Author):

My main concern is that large parts of the newly added text and figures are very confusing to me, to the point that I cannot understand what the authors mean after several attempts. Unfortunately, in my opinion the manuscript has not been improved following revision.

AUTHORS: We thank the reviewer for taking the time to go through a second detailed revision. As noted, many of the changes under discussion were specifically requested by Reviewer 1, who has approved the changes and believe they nicely convey the results. This gives us reassurance that the content was clear. We respond to each comment in turn, aiming primarily to ensure our analysis is as clear and transparent as possible.

The subsection "Core sets of ASVs determined convergence towards the attractors" relies on the "propensity", whose mathematical definition is something like the log-fold change in the probability of observing an ASV in a "convergent to Class 1" trajectory compared to the probability of observing that ASV in any trajectory. This definition is crucial for all of the subsequent analysis in the subsection and Figures 5 and 6, but its mathematical definition and graphical description (Figure 5a) are very difficult to parse, and I do not believe a typical reader of Nature Communications would understand what it means. The remaining content in this subsection is overly detailed (and more likely should belong in the supplement than the main text) and fails to convincingly summarize their later claim of "identifying two tipping points" that determine the compositional outcome of the replicates. Figure 6 is not even mentioned in the main text.

AUTHORS: We have revised this section to include a lay description of the definition of propensity. We have also removed some details, as suggested by the reviewer. We have also made reorganised the manuscript to make the logic of our analysis more clear, presenting all the results related to the compositional analysis first (including the new section) and then the functional analysis. We believe that the presentation is now more natural for a general audience and we thank the reviewer for their suggestions.

Regarding Figure 6: the figure is associated with the Methods rather than the Main Text, with the lay version (Fig 4A) more suitable for the main text. We have followed the journal guidelines in numbering the figure in the order in which they appear (with figures from the

methods numbered last). For clarity, we now refer the reader to the Methods in the Main Text, which is where Fig. 6 is cited.

I have a few other minor points that I hope the authors find useful:

+ line 114: The sentence mentioning the "small and significant Root Mean Square Deviation" does not communicate the result that the rigid-body transformation works better with the real data than for a null model in which the starting communities are shuffled, which I think is what the authors mean to say

AUTHORS: We clarified this sentence to communicate the result more clearly.

+ line 130: should be "communities'" (with an apostrophe)

AUTHORS: We could not see a grammatical mistake here. We suggest leaving to the copy editors in this instance.

+ line 144: while OTU vs ASV certainly introduces some discrepancy, unsupervised clustering must also play some role

AUTHORS: The communities and underlying data were the same and the clustering method and criteria to find the optimal partitioning in clusters were also the same (and both methods are deterministic). Therefore the discrepancy we believe can only come from the use of OTUs vs. ASVs.

+ line 146: grammatically, should be "providing strong evidence of selection"

AUTHORS: Corrected. Thank you.

+ line 452: How did the authors "bootstrap the data 1000 times"? There are many ways to bootstrap an ASV table.

AUTHORS: We now indicated that the bootstrapped ASV table is created by sampling communities with repetition.

+ line 453: The propensity can be negative, yet the authors specify that they "require" the 95% CI to be positive. Why?

AUTHORS: We were interested in identifying ASVs with a greater probability for being in a specific trajectory than for being in any trajectory. We therefore need to find positive propensities, and the CI needs to be well above zero to reject the hypothesis that the propensity is zero (i.e. no propensity). The reasoning is similar to demanding a significantly positive correlation coefficient, where the bootstrapped distribution of the correlation coefficient should also be above zero to consider the correlation to be significantly positive. We have clarified this sentence in the text.

+ I am still not sure how to interpret Figure 4c, and I also find the final sentence of the caption of Figure 4 confusing, since it seems that the green and brown points in Fig 4c are

distinguished by the minimum distance to centroids, and are not strongly affected by the mean distance to borders.

AUTHORS: We thank the reviewer for finding this typo, both terms "centroid" and "border" were mistakenly interchanged in the sentence.